# Revisiting the General Identifiability Problem

**Yaroslav Kivva**[1]       **Ehsan Mokhtarian**[1]       **Jalal Etesami**[1]       **Negar Kiyavash**[1,2]

[1]School of Computer and Communication Sciences, EPFL, Lausanne, Switzerland
[2]College of Management of Technology, EPFL, Lausanne, Switzerland

## Abstract

We revisit the problem of general identifiability originally introduced in [Lee et al., 2019] for causal inference and note that it is necessary to add positivity assumption of observational distribution to the original definition of the problem. We show that without such an assumption the rules of do-calculus and consequently the proposed algorithm in [Lee et al., 2019] are not sound. Moreover, adding the assumption will cause the completeness proof in [Lee et al., 2019] to fail. Under positivity assumption, we present a new algorithm that is provably both sound and complete. A nice property of this new algorithm is that it establishes a connection between general identifiability and classical identifiability by Pearl [1995] through decomposing the general identifiability problem into a series of classical identifiability sub-problems.

## 1 INTRODUCTION

Causal effect identification (or ID for short) problem, a central concern in causal inference, pertains to whether, given a causal graph, an interventional distribution can be uniquely computed from observational distribution [Pearl, 2009]. When all the variables in the system are observable, Pearl's do-calculus (a collection of three rules) allows determining whether a causal effect is identifiable [Pearl, 1995]. Moreover, it was shown that Pearl's do-calculus is both sound and complete for ID problem [Shpitser and Pearl, 2006a, Huang and Valtorta, 2008].

In the classical setting of ID problem, both the causal graph and the observational distribution, denoted by $P(\mathbf{V})$ ($\mathbf{V}$ is the set of observed variables in the causal graph), are given. However, it is assumed that no extra information (such as interventional distribution) is available. Recently, several work in the literature relax these assumptions [Tikka et al., 2021, Shpitser and Pearl, 2006b, Bareinboim and Tian, 2015, Bareinboim and Pearl, 2014, Mokhtarian et al., 2022]. Before discussing these results, let us introduce a notion. We denote by $P_{\mathbf{x}}(\mathbf{Y})$ the distribution of a set of variables $\mathbf{Y}$ resulting from intervening on another set of variables $\mathbf{X}$. Bareinboim and Pearl [2012] introduced the z-identification problem (or zID for short) in which for a fixed set $\mathbf{Z} \subseteq \mathbf{V}$, given a set of interventional distributions of the form $\{P_{\mathbf{z}'}(\mathbf{V}) : \forall \mathbf{Z}' \subseteq \mathbf{Z}\}$, one asks whether $P_{\mathbf{x}}(\mathbf{Y})$ is identifiable. Note that the observational distribution $P(\mathbf{V})$ always belongs to the set of available distributions. Furthermore, the form of given interventional distributions is restrictive. Lee et al. [2019] generalized zID and proposed so-called general identifiability problem (or gID for short). In the gID, observational distribution is not necessarily given but instead we have access to $\{P_{\mathbf{z}_i}(\mathbf{V})\}_{i=0}^{m}$ for some subsets $\{\mathbf{Z}_i\}_{i=0}^{m}$ of observed variables. When one of $\mathbf{Z}_i$s is an empty set, we have access to $P(\mathbf{V})$.

We give formal definitions of identifiability (Definition 3) and general identifiability (Definition 5) in Section 2. An important contribution of this paper is to add an assumption on the positivity of the observational distribution in the definition of general identifiability, i.e., $P(\mathbf{v}) > 0$ for all the realizations of observed variables. As we shall discuss in detail in Section 3, this assumption, or at least a relaxed version of it, is crucial. More specifically, do-calculus-based methods are no longer sound for the ID problem if we ignore the positivity assumption. In other words, there exist causal graphs with non-positive distribution $P(\mathbf{V})$ such that do-calculus would claim a causal effect is identifiable while it cannot be uniquely computed from mere observational distribution. Violation of the positivity assumption can happen in practice. For instance, some empirical distributions would be zero when the observational data is not large enough. An even more important reason for including the positivity assumption is that without it, the proposed algorithm in the original gID in [Lee et al., 2019] is not sound. Furthermore, as we shall discuss in Section 3, the proof of completeness in [Lee et al., 2019] relies on building two models that have

*Accepted for the 38th Conference on Uncertainty in Artificial Intelligence* (UAI 2022).

zero probabilities for certain realizations of observed variables. Therefore, unfortunately, simply adding the positivity assumption to the definition of general identifiability (g-identifiability) will fail the proof technique in [Lee et al., 2019] for the completeness of their proposed algorithm. On the other hand, ignoring the positivity assumption makes the soundness of their algorithm incorrect.

In summary, our main contributions are as follows. We redefine the g-identifiability by adding the positivity assumption of observational distribution (Definition 5). We show in Section 3 that this assumption is essential for the gID problem. We then provide a sound and complete algorithm for the gID problem (Algorithm 2). A nice property of our algorithm is that it establishes a connection between gID and classical ID by showing that gID can be reduced to solving a series of ID problems (Theorem 1).

## 2 PRELIMINARIES

### 2.1 TERMINOLOGY

Throughout the paper, we denote random variables by capital letters (e.g., $X$), their realizations by small letters (e.g., $x$), and sets by bold letters (e.g., $\mathbf{X}$ or $\mathbf{x}$). We use $\mathfrak{X}_X$ to denote the domain of random variable $X$ and $\mathfrak{X}_{\mathbf{X}}$ to denote the Cartesian product of the domains of all the variables in set $\mathbf{X}$, i.e., $\prod_{X \in \mathbf{X}} \mathfrak{X}_X$. For integer numbers $a \leq b$, we use $[a : b]$ to denote $\{a, a+1, \cdots, b\}$.

Suppose $\mathcal{G} = (\mathbf{V} \cup \mathbf{U}, \mathbf{E})$ is a directed acyclic graph (DAG) over vertex set $\mathbf{V} \cup \mathbf{U}$, where $\mathbf{V}$ and $\mathbf{U}$ represent the set of observed and unobserved variables, respectively. For each edge $(X, Y) \in \mathbf{E}$, $X$ is called a parent of $Y$, and $Y$ is called a child of $X$. Vertex $X$ is an ancestor of $Y$ in $\mathcal{G}$ if a directed path exists from $X$ to $Y$ in $\mathcal{G}$. Note that $X$ is an ancestor of itself. $Pa_{\mathcal{G}}(X)$, $Ch_{\mathcal{G}}(X)$, and $Anc_{\mathcal{G}}(X)$ denote the set of parents, children, and ancestors of $X$ in $\mathcal{G}$, respectively. These notations are also used for a set of vertices. In this case, they refer to the union over the set elements. For instance, $Pa_{\mathcal{G}}(\mathbf{X}) = \bigcup_{X \in \mathbf{X}} Pa_{\mathcal{G}}(X)$. We assume $\mathcal{G}$ is semi-Markovian, that is for each $U \in \mathbf{U}$, $Pa_{\mathcal{G}}(U) = \varnothing$ and $|Ch_{\mathcal{G}}(U)| = 2$. Note that this is not a restrictive assumption as there exists an equivalency for identifiability in DAGs and semi-Markovian DAGs [Huang and Valtorta, 2006].

Structural Equation Models (SEMs) are used to model causal systems [Pearl, 2009]. $\mathcal{G}$ is a causal graph for SEM $\mathcal{M}$ if each $X \in \mathbf{V} \cup \mathbf{U}$ is generated as $f_X(Pa_{\mathcal{G}}(X), \epsilon_X)$, where $\{\epsilon_X : X \in \mathbf{V}\}$ is a set of mutually independent exogenous random variables. We denote by $P^{\mathcal{M}}(\cdot)$ the joint distribution of the variables in $\mathcal{M}$ and drop the superscript $\mathcal{M}$ when it is clear from the context. Markov factorization property implies that $P^{\mathcal{M}}(\cdot)$ can get factorized as

$$P(\mathbf{v}) = \sum_{\mathbf{U}} \prod_{X \in \mathbf{V}} P(x|Pa_{\mathcal{G}}(X)) \prod_{U \in \mathbf{U}} P(u), \quad (1)$$

where $\sum_{\mathbf{U}}$ denotes the marginalization over $\mathbf{U}$.

**Definition 1.** $\mathbb{M}(\mathcal{G})$ *denotes the set of SEMs with causal graph* $\mathcal{G}$. $\mathbb{M}^+(\mathcal{G})$ *denotes the set of SEMs* $\mathcal{M} \in \mathbb{M}(\mathcal{G})$ *such that* $P^{\mathcal{M}}(\mathbf{v}) > 0$ *for each* $\mathbf{v} \in \mathfrak{X}_{\mathbf{V}}$.

For $\mathbf{X} \subseteq \mathbf{V}$ and $\mathbf{x} \in \mathfrak{X}_{\mathbf{X}}$, the intervention $do(\mathbf{X} = \mathbf{x})$ converts $\mathcal{M}$ to a new SEM where the equations of $\mathbf{X}$ in $\mathcal{M}$ are replaced by the constants in $\mathbf{x}$. We denote by $P_{\mathbf{x}}(\cdot)$ the corresponding post interventional distribution.

**Remark 1.** *For three disjoint subsets* $\mathbf{X}, \mathbf{Y}, \mathbf{W}$ *of* $\mathbf{V}$, *if* $\mathcal{M} \in \mathbb{M}^+(\mathcal{G})$, *then* $P_{\mathbf{x}}^{\mathcal{M}}(\mathbf{y} \mid \mathbf{w}) > 0$ *for any* $\mathbf{x} \in \mathfrak{X}_{\mathbf{X}}$, $\mathbf{y} \in \mathfrak{X}_{\mathbf{Y}}$, *and* $\mathbf{w} \in \mathfrak{X}_{\mathbf{W}}$.

For $\mathbf{v} \in \mathfrak{X}_{\mathbf{V}}$ and $\mathbf{S} \subseteq \mathbf{V}$, we define $Q[\mathbf{S}](\cdot)$ by

$$Q[\mathbf{S}](\mathbf{v}) := P_{\mathbf{v} \setminus \mathbf{s}}(\mathbf{s}). \quad (2)$$

Similar to Equation (1), $Q[\mathbf{S}]$ can get factorized as

$$Q[\mathbf{S}](\mathbf{v}) = \sum_{\mathbf{U}} \prod_{S \in \mathbf{S}} P(s|Pa_{\mathcal{G}}(S)) \prod_{U \in \mathbf{U}} P(u). \quad (3)$$

For $\mathbf{X} \subseteq \mathbf{V}$, $\mathcal{G}[\mathbf{X}]$ denotes the inducing subgraph of $\mathcal{G}$ over $\mathbf{X}$ and the unobserved variables with both children in $\mathbf{X}$. Note that $\mathcal{G}$ is semi-Markovian. Furthermore, we denote by $\mathcal{G}_{\mathbf{X}}$ the partially directed graph over $\mathbf{X}$ obtained by removing unobserved variables of $\mathcal{G}[\mathbf{X}]$ and replacing them by bidirected edges.

**Definition 2 (c-component, c-forest).** *For* $\mathbf{X} \subseteq \mathbf{V}$, *confounded components or c-components of* $\mathbf{X}$ *are the connected components of the graph obtained by only the bidirected edges of* $\mathcal{G}_{\mathbf{X}}$. *Also, a subgraph of* $\mathcal{G}_{\mathbf{V}}$ *is called a single c-component if its bidirected edges form a connected graph. Suppose* $\mathcal{H}$ *is a subgraph of* $\mathcal{G}$ *over observed vertices* $\mathbf{X}$. *The root set of* $\mathcal{H}$ *is the maximal subset of* $\mathbf{X}$ *with no children in* $\mathcal{H}$. $\mathcal{H}$ *is called* $\mathbf{R}$*-rooted c-forest if* $\mathbf{R}$ *is the root set of* $\mathcal{H}$, $\mathcal{H}_{\mathbf{X}}$ *is a single c-component, and each node in* $\mathbf{X}$ *has at most one child in* $\mathcal{H}$.

**Example 1:** Consider the causal graph $\mathcal{G}$ in Figure 1a, where $\mathbf{V} = \{X_1, X_2, Y_1, Y_2\}$ and $\mathbf{U} = \{U_1, U_2\}$. $\mathcal{G}_{\mathbf{V}}$ is depicted in Figure 1b. The c-components of $\mathbf{V}$ are $\{X_1, X_2\}$ and $\{Y_1, Y_2\}$. Figure 1c depicts the inducing subgraph of $\mathcal{G}$ over $\{X_1, X_2\}$ and $\mathcal{G}_{\{X_1, X_2\}}$ is depicted in Figure 1d. Herein, $\mathcal{G}[\{X_1, X_2\}]$ is $\{X_1, X_2\}$-rooted c-forest since $\mathcal{G}_{\{X_1, X_2\}}$ is single c-component.

### 2.2 IDENTIFIABILITY

The goal in the *identifiability* problem is to understand whether a post-interventional distribution can be uniquely computed from observational distribution $P(\mathbf{V})$, given the causal graph [Pearl, 2009].

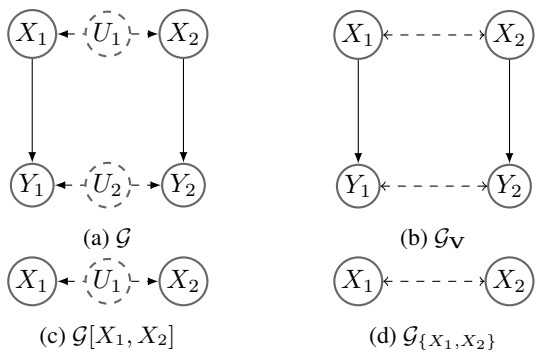

(a) $\mathcal{G}$           (b) $\mathcal{G}_{\mathbf{V}}$

(c) $\mathcal{G}[X_1, X_2]$       (d) $\mathcal{G}_{\{X_1, X_2\}}$

Figure 1: An example for a causal DAG $\mathcal{G}$ over observed variables $\mathbf{V} = \{X_1, X_2, Y_1, Y_2\}$ and unobserved variables $\mathbf{U} = \{U_1, U_2\}$.

**Definition 3** (identifiability). *Suppose $\mathbf{X}$ and $\mathbf{Y}$ are two disjoint subsets of $\mathbf{V}$. The causal effect of $\mathbf{X}$ on $\mathbf{Y}$ is said to be identifiable from $\mathcal{G}$ if for any $\mathbf{x} \in \mathfrak{X}_{\mathbf{X}}$ and $\mathbf{y} \in \mathfrak{X}_{\mathbf{Y}}$, $P_{\mathbf{x}}^{\mathcal{M}}(\mathbf{y})$ is uniquely computable from $P^{\mathcal{M}}(\mathbf{V})$ in any SEM $\mathcal{M} \in \mathbb{M}^+(\mathcal{G})$. Also, $Q[\mathbf{Y}]$ is said to be identifiable from $\mathcal{G}$ if the causal effect of $\mathbf{V} \setminus \mathbf{Y}$ on $\mathbf{Y}$ is identifiable from $\mathcal{G}$.*

Huang and Valtorta [2008] showed that identifiability of a causal effect is equivalent to identifiability of a specific $Q[\cdot]$.

**Proposition 1** (Huang and Valtorta [2008]). *The causal effect of $\mathbf{X}$ on $\mathbf{Y}$ is identifiable from $\mathcal{G}$ if and only if $Q[Anc_{\mathcal{G}_{\mathbf{V} \setminus \mathbf{x}}}(\mathbf{Y})]$ is identifiable from $\mathcal{G}$.*

For a subset $\mathbf{S}$ of observed nodes, [Tian and Pearl, 2003] showed that identifiability of a $Q[\mathbf{S}]$ is equivalent to identifiability of all its c-components.

**Proposition 2** (Tian and Pearl [2003]). *Suppose $\mathbf{S}_1, \cdots, \mathbf{S}_l$ are the c-components of $\mathbf{S} \subseteq \mathbf{V}$. $Q[\mathbf{S}]$ is identifiable from $\mathcal{G}$ if and only if $Q[\mathbf{S}_i]$ is identifiable from $\mathcal{G}$ for all $i \in [1 : l]$.*

Based on Propositions 1 and 2, [Tian and Pearl, 2003] proposed an algorithm that for two disjoint subsets $\mathbf{X}$ and $\mathbf{Y}$ checks the identifiability of the causal effect of $\mathbf{X}$ on $\mathbf{Y}$ from observational distribution given the causal graph $\mathcal{G}$. As we will use their algorithm as a subroutine in our algorithm for g-identifiability, we present their method in Algorithm 1. In this algorithm, function **ID_Single** determines whether $Q[\mathbf{S}]$ is identifiable from $\mathcal{G}$ when $\mathbf{S}$ is a single c-component. More precisely, this function starts from $\mathbf{Y} = \mathbf{V}$ and at each step, it decreases $\mathbf{Y}$ such that both $Q[\mathbf{Y}]$ remains identifiable from $\mathcal{G}$ and $\mathbf{S} \subseteq \mathbf{Y}$. If this procedure can reduce $\mathbf{Y}$ to $\mathbf{S}$, then $Q[\mathbf{S}]$ is identifiable otherwise, $Q[\mathbf{S}]$ is not identifiable. This algorithm is both sound and complete [Shpitser and Pearl, 2006a, Huang and Valtorta, 2008].

## 2.3 GENERAL IDENTIFIABILITY

In the previous section, we explained the classical identifiability problem which determines whether a causal effect is

---

**Algorithm 1:** Identifiability
1: **Function ID**($\mathbf{X}, \mathbf{Y}, \mathcal{G}$)
2:   **Output:** True, if the causal effect of $\mathbf{X}$ on $\mathbf{Y}$ is identifiable from $\mathcal{G}$.
3:   $\mathbf{S} \leftarrow Anc_{\mathcal{G}_{\mathbf{V} \setminus \mathbf{x}}}(\mathbf{Y})$
4:   $\{\mathbf{S}_1, \ldots, \mathbf{S}_l\} \leftarrow$ c-components of $\mathbf{S}$
5:   **for** $i$ from 1 to $l$ **do**
6:     **if ID_Single**($\mathbf{S}_i, \mathcal{G}$) = False **then**
7:       **Return** False
8:   **Return** True

---

1: **Function ID_Single**($\mathbf{S}, \mathcal{G}$)
2:   **Output:** True, if $Q[\mathbf{S}]$ is identifiable from $\mathcal{G}$, where $\mathbf{S}$ is a single c-component.
3:   $\mathbf{Y} \leftarrow \mathbf{V}$
4:   **while** $\mathbf{Y} \neq \mathbf{S}$ **do**
5:     $\mathbf{A} \leftarrow Anc_{\mathcal{G}_{\mathbf{Y}}}(\mathbf{S})$
6:     $\mathbf{Y}_{new} \leftarrow$ The c-component of $\mathbf{A}$ that contains $\mathbf{S}$
7:     **if** $\mathbf{Y}_{new} = \mathbf{Y}$ **then**
8:       **Return** False
9:     **else**
10:       $\mathbf{Y} \leftarrow \mathbf{Y}_{new}$
11: **Return** True

---

identifiable from observational distribution given the causal graph. As we discussed earlier, in many problems of interest, the goal is to identify a causal effect from a set of both observational and interventional distributions given a causal graph. A variant of this problem was defined by Lee et al. [2019] under the name g-identifiability.

**Definition 4** (g-identifiability in [Lee et al., 2019]). *Let $\mathbf{X}, \mathbf{Y}$ be disjoint subsets of $\mathbf{V}$, $\mathbb{Z} = \{\mathbf{Z}_i\}_{i=0}^m$ be a collection of subsets of $\mathbf{V}$, and let $\mathcal{G}$ be a causal diagram. $P_{\mathbf{x}}(\mathbf{y})$ is said to be g-identifiable from $\mathbb{Z}$ in $\mathcal{G}$, if $P_{\mathbf{x}}(\mathbf{y})$ is uniquely computable from distributions $\{P(\mathbf{V}|do(\mathbf{z}))\}_{\mathbf{Z} \in \mathbb{Z}, \mathbf{z} \in \mathfrak{X}_{\mathbf{z}}}$ in any causal model which induces $\mathcal{G}$.*

Note that the causal model in this definition belongs to $\mathbb{M}(\mathcal{G})$. However, as we shall discuss in Section 3, it is crucial to assume that the causal model is positive, i.e., it belongs to $\mathbb{M}^+(\mathcal{G})$. Therefore, we modify the above definition as follows.

**Definition 5** (g-identifiability). *Suppose $\mathbb{A} = \{\mathbf{A}_i\}_{i=0}^m$ is a collection of subsets of $\mathbf{V}$ and $\mathbf{X}, \mathbf{Y}$ are two disjoint subsets of $\mathbf{V}$. The causal effect of $\mathbf{X}$ on $\mathbf{Y}$ is said to be g-identifiable from $(\mathbb{A}, \mathcal{G})$ if for any $\mathbf{x} \in \mathfrak{X}_{\mathbf{X}}$ and $\mathbf{y} \in \mathfrak{X}_{\mathbf{Y}}$, $P_{\mathbf{x}}^{\mathcal{M}}(\mathbf{y})$ is uniquely computable from the set of distributions $\{Q[\mathbf{A}_i]\}_{i=0}^m$ in any SEM $\mathcal{M} \in \mathbb{M}^+(\mathcal{G})$. Also, $Q[\mathbf{Y}]$ is said to be g-identifiable from $(\mathbb{A}, \mathcal{G})$ if the causal effect of $\mathbf{V} \setminus \mathbf{Y}$ on $\mathbf{Y}$ is g-identifiable from $(\mathbb{A}, \mathcal{G})$.*

Note that knowing $P(\mathbf{V}|do(\mathbf{Z}))$ for some subset $\mathbf{Z} \subseteq \mathbf{V}$ is equivalent to knowing $Q[\mathbf{V} \setminus \mathbf{Z}]$, and therefore, by setting

$\mathbf{A}_i = \mathbf{V} \setminus \mathbf{Z}_i$, the two aforementioned definitions are the same except for the positivity assumption. For the remainder of this paper, we use Definition 5 for g-identifiability.

# 3 ON THE POSITIVITY ASSUMPTION IN G-IDENTIFIABILITY

In our definition of g-identifiability and the classical definition of identifiability (Definitions 5 and 3), only SEMs that belong to $\mathbb{M}^+(\mathcal{G})$ instead of $\mathbb{M}(\mathcal{G})$ are considered [Huang and Valtorta, 2008, Shpitser and Pearl, 2006a]. That is, SEMs with positive probabilities for any realization $\mathbf{v} \in \mathfrak{X}_\mathbf{V}$. In this section, we discuss why this assumption is crucial by showing that ignoring positivity leads to wrong conclusions. As a consequence, since Lee et al. [2019] presented the soundness and completeness of their algorithm for g-identifiability, ignoring the positivity assumption, we discuss how after imposing the assumption, their results are no longer valid. We further show that this issue cannot be fixed by the relaxed version of the positivity assumption introduced by [Shpitser and Pearl, 2006a]. After this discussion, we present a new algorithm in the next section for g-identifiability and prove its soundness and completeness under the positivity assumption.

## 3.1 SOUNDNESS REQUIRES POSITIVITY

The following example shows that do-calculus-based methods (e.g., Algorithm 1) are no longer sound for the ID problem ignoring the positivity assumption.

**Example 2:** Consider again the causal graph in Figure 1. Herein, do-calculus-based methods (e.g., Algorithm 1) would report that the causal effect of $\mathbf{X} = \{X_1, X_2\}$ on $\mathbf{Y} = \{Y_1, Y_2\}$ is identifiable given $\mathcal{G}$. However, by ignoring the positivity assumption, we can introduce two SEMs $\mathcal{M}_1$ and $\mathcal{M}_2$ in $\mathbb{M}(\mathcal{G})$ that have the same observational distribution but result in two different post-interventional distributions after intervening on $\{X_1, X_2\}$. This clearly contradicts with the identifiability of $P_{x_1, x_2}(y_1, y_2)$.

All variables in both models are binary. Also, for both models and $i \in \{1, 2\}$, we define $P(U_i = 0) = P(U_i = 1) = 0.5$ and $X_i = U_i$. In model $\mathcal{M}_1$, we define $Y_1, Y_2$ to have the following conditional distributions:

$$P^{\mathcal{M}_1}(y_1 \mid u_2, x_1) = \frac{1}{3}\mathbb{1}_{y_1=u_2} + \frac{2}{3}\mathbb{1}_{y_1 \neq u_2},$$
$$P^{\mathcal{M}_1}(y_2 \mid u_2, x_2) = \frac{1}{3}\mathbb{1}_{y_2=(u_2 \oplus x_2)} + \frac{2}{3}\mathbb{1}_{y_2 \neq (u_2 \oplus x_2)},$$

where $\mathbb{1}_A$ is the indicator function which is one whenever the statement in $A$ is true and is zero otherwise. For model

$\mathcal{M}_2$, we define the conditional distributions of $Y_1, Y_2$ as

$$P^{\mathcal{M}_2}(y_1 \mid u_2, x_1) = \frac{2}{3}\mathbb{1}_{y_1=(u_2 \oplus x_1)} + \frac{1}{3}\mathbb{1}_{y_1 \neq (u_2 \oplus x_1)},$$
$$P^{\mathcal{M}_2}(y_2 | u_2, x_2) = \frac{2}{3}\mathbb{1}_{y_2=u_2} + \frac{1}{3}\mathbb{1}_{y_2 \neq u_2}.$$

It is straightforward to see that for any realizations $(x_1, x_2, y_1, y_2) \in \mathfrak{X}_\mathbf{V}$, we have

$$P^{\mathcal{M}_1}(x_1, x_2, y_1, y_2) = P^{\mathcal{M}_2}(x_1, x_2, y_1, y_2).$$

However,

$$\frac{4}{9} = P^{\mathcal{M}_1}_{x_1=0, x_2=1}(Y_1 = 0, Y_2 = 0)$$
$$\neq P^{\mathcal{M}_2}_{x_1=0, x_2=1}(Y_1 = 0, Y_2 = 0) = \frac{5}{9}.$$

Note that $\mathcal{M}_1$ and $\mathcal{M}_2$ do not belong to $\mathbb{M}^+(\mathcal{G})$, since $P(x_1 = 0, x_2 = 1, y_1, y_2) = 0$ for any $y_1 \in \mathfrak{X}_{Y_1}$ and $y_2 \in \mathfrak{X}_{Y_2}$. This example shows that if we use $\mathbb{M}(\mathcal{G})$ instead of $\mathbb{M}^+(\mathcal{G})$ in Definition 3, the causal effect of $\mathbf{X}$ on $\mathbf{Y}$ is not identifiable from $\mathcal{G}$, and therefore, do-calculus-based methods such as the proposed algorithm in Lee et al. [2019] are not sound. Specifically, the proposed algorithm in Lee et al. [2019] suggests the causal effect in this example is g-identifiable and returns the following expression:

$$P_{x_1, x_2}(y_1, , y_2) = P(y_1|x_1, x_2)P(y_2|y_1, x_2, x_1).$$

This expression is not well-defined for all realizations ignoring the positivity assumption because for some realizations $P(x_1, x_2)$ is zero which means the conditional distribution $P(y_1|x_1, x_2)$ is not well-defined. Thus, the algorithm in Lee et al. [2019] *is not sound*.

Next, we discuss the g-identifiability in Lee et al. [2019] and show that the completeness result provided in that work relies on two models in $\mathbb{M}(\mathcal{G})$ that violate the positivity assumption.

## 3.2 COMPLETENESS

[Lee et al., 2019] presented necessary and sufficient conditions to determine if a causal effect $P_\mathbf{x}(\mathbf{y})$ is g-identifiable w.r.t. the Definition 4. To prove that their proposed conditions are necessary for g-identifiability, they construct two models $\mathcal{M}_1$ and $\mathcal{M}_2$ such that the available distributions in the definition of the problem are the same for both models yet $P^{\mathcal{M}_1}_\mathbf{x}(\mathbf{y}) \neq P^{\mathcal{M}_2}_\mathbf{x}(\mathbf{y})$. The issue here is that they constructed their models ignoring the positivity assumption, allowing for zero probability for some realizations. In fact, having zero probabilities in their model is essential for the proof. For instance, Lemma 3 in Lee et al. [2019] states that under certain conditions, there is an observed variable $R \in \mathbf{V}$ such that it takes value zero in both their models with probability one. In other words, the probability of $R$

not being zero is zero (see Appendix 1.2 for more details.) This shows that adding the positivity assumption to the definition of gID will fail the proof technique in [Lee et al., 2019] for the completeness of their proposed algorithm.

It is noteworthy to mention that an alternative positivity assumption is introduced by Shpitser and Pearl [2006a]. Below, we describe this assumption and discuss that the models introduced in Lee et al. [2019] also violate this assumption.

## 3.3 RELAXED POSITIVITY ASSUMPTION

Shpitser and Pearl [2006a] show that in the ID problem of a causal effect $P_{\mathbf{x}}(\mathbf{y})$, one can relax the positivity constraint $P(\mathbf{V}) > 0$ to $P(\mathbf{X} | (Pa_{\mathcal{G}}(\mathbf{X}) \cap \mathbf{V}) \setminus \mathbf{X}) > 0$. They show that the rules of do-calculus are sound under the relaxed positivity assumption. However, as we mentioned, even the relaxed constraint does not hold for the constructed models in Lee et al. [2019]. More precisely, consider the causal graph $\mathcal{G}$ in Figure 2 which is brought here from Lee et al. [2019]. Assume that we are interested in g-identifying the causal effect $Q[R]$ from $\mathbb{Z} = \{\varnothing\}$, i.e., from mere observational distribution $P(\mathbf{V})$, w.r.t. Definition 4. In this case, $\mathbf{X} = \{T_1, T_2, T_3\}$ and therefore:

$$P(\mathbf{X} | (Pa_{\mathcal{G}}(\mathbf{X}) \cap \mathbf{V}) \setminus \mathbf{X}) = P(T_1, T_2, T_3).$$

The result in Lee et al. [2019] implies that the causal effect $Q[R]$ is not g-identifiable given the causal graph $\mathcal{G}$ in Figure 2. To prove the non g-identifiability, Lee et al. [2019] constructed two models $\mathcal{M}_1$ and $\mathcal{M}_2$ that impose similar observational distributions, i.e., $P^{\mathcal{M}_1}(\mathbf{V}) = P^{\mathcal{M}_2}(\mathbf{V})$, while the causal effect $Q[R]$ under these two models are not the same for at least one realization. Next, we present these two models and show that they violate the positivity assumption claimed in Shpitser and Pearl [2006a], i.e., $P(T_1, T_2, T_3)$ is zero for certain realizations of $\{T_1, T_2, T_3\}$.

By the construction in Lee et al. [2019], variables $T_3, U_1, U_2, U_3$ are binary variables and $T_1, T_2$ are binary vectors of length two. For both models, all unobserved variables are defined to be binary with uniform distribution, and the observed variables $T_1, T_2, T_3$ are defined as follows.

$$
\begin{aligned}
T_3 &= U_2 \oplus U_3, \\
T_{2,1} &= T_3, \quad T_{2,2} = U_1, \\
T_{1,1} &= T_{2,1} \oplus U_2, \quad T_{1,2} = T_{2,2}.
\end{aligned}
$$

In model $\mathcal{M}_1$, variable $R$ is defined as

$$R = \mathbb{1}_{T_{1,1}=0} \wedge \mathbb{1}_{T_{1,2}=0} \wedge \mathbb{1}_{U_3=1} \wedge \mathbb{1}_{U_1=1},$$

and in model $\mathcal{M}_2$, it is defined to be zero, i.e., $R = 0$.

Given the above models, it is clear that the probability $P(t_1, t_2, t_3)$ is equal to zero whenever $t_{2,1} \neq t_3$, and therefore, the relaxed positivity constraint $P(T_1, T_2, T_3) > 0$

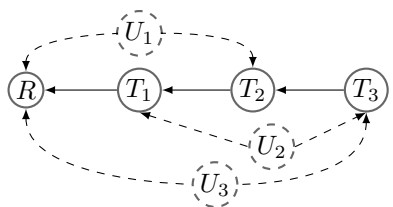

Figure 2: A causal graph of [Lee et al., 2019] that shows the violation of relaxed positivity assumption in constructed models of [Lee et al., 2019].

does not hold for the models in Lee et al. [2019]. See Appendix 1 for more details.

To summarize, in this section, our goal was to prove the importance of positivity assumption in both classical ID and its generalization gID. We did so by showing that the rules of *do-calculus* and consequently the proposed algorithm in Lee et al. [2019] are not sound without the positivity assumption. Moreover, we discussed that the completeness proof in Lee et al. [2019] only holds when there is no positivity assumption. This motivates our work to revisit the gID problem by including the positivity assumption in the definition of gID and presenting a new algorithm that is provably sound and complete.

## 4 AN ALGORITHM FOR GID

In this section, we propose an algorithm for gID from $(\mathbb{A}, \mathcal{G})$, where $\mathbb{A} = \{\mathbf{A}_i\}_{i=0}^m$ is a collection of subsets of $\mathbf{V}$. To this end, we first extend Propositions 1 and 2 from identifiability to g-identifiability.

**Proposition 3.** *Let $\mathbf{X}$ and $\mathbf{Y}$ be two disjoint subsets of $\mathbf{V}$. The causal effect of $\mathbf{X}$ on $\mathbf{Y}$ is g-identifiable from $(\mathbb{A}, \mathcal{G})$ if and only if $Q[Anc_{\mathcal{G}_{\mathbf{V} \setminus \mathbf{x}}}(\mathbf{Y})]$ is g-identifiable from $(\mathbb{A}, \mathcal{G})$.*

**Proposition 4.** *Suppose $\mathbf{S}_1, \cdots, \mathbf{S}_l$ are the c-components of $\mathbf{S} \subseteq \mathbf{V}$. $Q[\mathbf{S}]$ is g-identifiable from $(\mathbb{A}, \mathcal{G})$ if and only if $Q[\mathbf{S}_i]$ is g-identifiable from $(\mathbb{A}, \mathcal{G})$ for all $i \in [1:l]$.*

Proofs are provided in Appendix 2. Proposition 3 allows us to solve the gID problem for $P_{\mathbf{x}}(\mathbf{Y})$ by solving an equivalent problem for $Q[\mathbf{S}]$, where $\mathbf{S}$ is given in the same proposition. Proposition 4 shows that the g-identifiability of $Q[\mathbf{S}]$ from $(\mathbb{A}, \mathcal{G})$ is equivalent to g-identifiability of its single c-components. The following result provides a method for solving g-identifiability of $Q[\mathbf{S}]$ when $\mathbf{S}$ is a single c-component.

**Theorem 1.** *Suppose $\mathbf{S} \subseteq \mathbf{V}$ is a single c-component. $Q[\mathbf{S}]$ is g-identifiable from $(\mathbb{A}, \mathcal{G})$ if and only if there exists $\mathbf{A} \in \mathbb{A}$ such that $\mathbf{S} \subseteq \mathbf{A}$ and $Q[\mathbf{S}]$ is identifiable from $\mathcal{G}[\mathbf{A}]$.*

A proof for Theorem 1 is provided in Section 5. Note that the equivalent condition provided in Theorem 1 is identifiability

**Algorithm 2:** g-identifiability

---
1: **Function** GID($\mathbf{X}, \mathbf{Y}, \mathbb{A} = \{\mathbf{A}_i\}_{i=0}^m, \mathcal{G}$)
2: **Output:** True, if the causal effect of $\mathbf{X}$ on $\mathbf{Y}$ is
    g-identifiable from $(\mathbb{A}, \mathcal{G})$.
3:  $\mathbf{S} \leftarrow Anc_{\mathcal{G}_{\mathbf{V}\setminus\mathbf{x}}}(\mathbf{Y})$
4:  $\{\mathbf{S}_1, \ldots, \mathbf{S}_l\} \leftarrow$ c-components of $\mathbf{S}$
5: **for** $i$ from 1 to $l$ **do**
6:    **if** GID_Single($\mathbf{S}_i, \mathbb{A} = \{\mathbf{A}_i\}_{i=0}^m, \mathcal{G}$) = False **then**
7:       **Return** False
8: **Return** True

---
1: **Function** GID_Single($\mathbf{S}, \mathbb{A} = \{\mathbf{A}_i\}_{i=0}^m, \mathcal{G}$)
2: **Output:** True, if $Q[\mathbf{S}]$ is identifiable from $(\mathbb{A}, \mathcal{G})$,
    where $\mathbf{S}$ is a single c-component.
3: **for** $i$ from 0 to $m$ **do**
4:    **if** $\mathbf{S} \subseteq \mathbf{A}_i$ and **ID_Single**($\mathbf{S}, \mathcal{G}[\mathbf{A}_i]$) = True **then**
5:       **Return** True
6: **Return** False

---

of a $Q[\cdot]$. This can be checked by function **ID_Single** in Algorithm 1. Therefore, when $\mathbf{S}$ is a single c-component, in order to check whether $Q[\mathbf{S}]$ is g-identifiable from $(\mathbb{A}, \mathcal{G})$, we need to check the identifiability of $Q[\mathbf{S}]$ from $\mathcal{G}[\mathbf{A}]$ for all $\mathbf{A} \in \mathbb{A}$ that $\mathbf{S} \subseteq \mathbf{A}$. Algorithm 2 summarizes the steps for solving g-identifiability of a causal effect given $(\mathbb{A}, \mathcal{G})$.

**Theorem 2.** *Algorithm 2 is sound and complete.*

*Proof.* It directly follows from Propositions 3 and 4 and Theorem 1. □

**Remark 2.** *Under the relaxed positivity assumption, the algorithm is still sound and complete because Algorithm 2 is based on the rules of do-calculus, and these rules are both sound and complete under the relaxed positivity assumption.*

Suppose Algorithm 2 determines that the causal effect of $\mathbf{X}$ on $\mathbf{Y}$ is g-identifiable from $(\mathbb{A}, \mathcal{G})$. Analogous to the method in Tian and Pearl [2003], we can derive a formula for $P_{\mathbf{x}}(\mathbf{Y})$ as follows. For each $\mathbf{S}_i \in \{\mathbf{S}_1, \cdots, \mathbf{S}_l\}$, we can derive a formula for $Q[\mathbf{S}_i]$ using **ID_Single** function in line 4 of **GID_Single**. This allows us to compute $Q[\mathbf{S}]$ using

$$Q[\mathbf{S}] = \prod_{i=1}^l Q[\mathbf{S}_i].$$

Finally, the expression for $P_{\mathbf{x}}(\mathbf{Y})$ will be

$$P_{\mathbf{x}}(\mathbf{Y}) = \sum_{\mathbf{S}\setminus\mathbf{Y}} Q[\mathbf{S}].$$

# 5   MAIN RESULT: THEOREM 1

In this section, we present the main steps of the proof of Theorem 1. The technical lemmas in this section are proved in Appendix 2.

**Sufficient part:** This part is straightforward: if $Q[\mathbf{S}]$ is identifiable from $\mathcal{G}[\mathbf{A}]$ for some $\mathbf{A} \in \mathbb{A}$ such that $\mathbf{S} \subseteq \mathbf{A}$, then $Q[\mathbf{S}]$ is uniquely computable from $Q[\mathbf{A}]$, and therefore, $Q[\mathbf{S}]$ is g-identifiable from $(\mathbb{A}, \mathcal{G})$.

**Necessary part:** Suppose $\mathbf{S}$ is a single c-component and $Q[\mathbf{S}]$ is not identifiable from $\mathcal{G}[\mathbf{A}_i]$ for all $\mathbf{A}_i \in \mathbb{A}$ such that $\mathbf{S} \subseteq \mathbf{A}_i$. We need to show that $Q[\mathbf{S}]$ is not g-identifiable from $(\mathbb{A}, \mathcal{G})$. Recall that $\mathbb{A} = \{\mathbf{A}_i\}_{i=0}^m$. To this end, we will introduce two SEMs $\mathcal{M}_1$ and $\mathcal{M}_2$ in $\mathbb{M}^+(\mathcal{G})$ such that for each $i \in [0:m]$ and any $\mathbf{v} \in \mathfrak{X}_{\mathbf{V}}$,

$$Q^{\mathcal{M}_1}[\mathbf{A}_i](\mathbf{v}) = Q^{\mathcal{M}_2}[\mathbf{A}_i](\mathbf{v}), \qquad (4)$$

but there exists $\mathbf{v}_0 \in \mathfrak{X}_{\mathbf{V}}$ such that

$$Q^{\mathcal{M}_1}[\mathbf{S}](\mathbf{v}_0) \neq Q^{\mathcal{M}_2}[\mathbf{S}](\mathbf{v}_0). \qquad (5)$$

This shows that $Q[\mathbf{S}]$ cannot be uniquely computed from $\{Q[\mathbf{A}_i]\}_{i=0}^m$.

For sake of space, we assume that there exists at least one $i \in [0, m]$ such that $\mathbf{S} \subset \mathbf{A}_i$. In this case, without loss of generality, we assume that there exists $k \in [0 : m]$ such that $\mathbf{S} \subset \mathbf{A}_i$ for $i \in [0 : k]$ and $\mathbf{S} \nsubseteq \mathbf{A}_i$ for $i \in [k + 1 : m]$. A proof for the case in which $\mathbf{S}$ is not a subset of $\mathbf{A}_i$ for all $i \in [0, m]$ is provided in Appendix 3.

We first modify $\mathcal{G}$ by deleting some nodes and edges and show that it is enough to prove Theorem 1 for the modified graph. Then, we provide our method for constructing $\mathcal{M}_2$ from $\mathcal{M}_1$ by introducing a system of linear equations.

**Graph modification:** Since $\mathbf{S}$ is single c-component, the bidirected edges in $\mathcal{G}_{\mathbf{S}}$ form a connected graph over $\mathbf{S}$. Let $\mathcal{F}^{\mathbf{S}}$ be a minimal (in terms of edges) spanning subgraph of $\mathcal{G}[\mathbf{S}]$ such that $\mathcal{F}_{\mathbf{S}}^{\mathbf{S}}$ is single c-component. Thus, $\mathcal{F}_{\mathbf{S}}^{\mathbf{S}}$ has no directed edges, and its bidirected edges form a spanning tree.

**Lemma 1** (Shpitser and Pearl [2006a]). *Suppose $\mathbf{S} \subseteq \mathbf{A} \subseteq \mathbf{V}$. $Q[\mathbf{S}]$ is not identifiable from $\mathcal{G}[\mathbf{A}]$ if and only if there exists at least one $\mathbf{S}$-rooted c-forest $\mathcal{F}$ with the set of observed variables $\mathbf{B}$ such that $\mathbf{S} \subsetneq \mathbf{B} \subseteq \mathbf{A}$, the bidirected edges of $\mathcal{F}_{\mathbf{B}}$ form a spanning tree, and the induced subgraph of $\mathcal{F}$ over $\mathbf{S}$ is $\mathcal{F}^{\mathbf{S}}$, i.e., $\mathcal{F}^{\mathbf{S}} = \mathcal{F}[\mathbf{S}]$.*

Recall that for each $i \in [0 : k]$, $\mathbf{S} \subset \mathbf{A}_i$ and $Q[\mathbf{S}]$ is not identifiable from $Q[\mathbf{A}_i]$. Hence, Lemma 1 implies that for each $i \in [0 : k]$, there exists a $\mathbf{S}$-rooted c-forest $\mathcal{F}_i$ over a set of observed variables $\mathbf{B}_i$ such that $\mathbf{S} \subsetneq \mathbf{B}_i \subseteq \mathbf{A}_i$, the bidirected edges of $(\mathcal{F}_i)_{\mathbf{S}}$ form a spanning tree, and $\mathcal{F}^{\mathbf{S}} = \mathcal{F}_i[\mathbf{S}]$. Next, we use $\{\mathcal{F}_i\}_{i=0}^k$ to modify $\mathcal{G}$.

We define $\mathcal{G}'$ to be the union of all the subgraphs in $\{\mathcal{F}_i\}_{i=0}^k$ with the observed variables $\mathbf{V}' := \bigcup_{i=0}^k \mathbf{B}_i$ and unobserved variables $\mathbf{U}'$. Furthermore, let $\mathbb{A}' := \{\mathbf{A}_i' := \mathbf{A}_i \cap \mathbf{V}'\}_{i=0}^m$. Because for each $i \in [0 : k]$, $\mathcal{F}_i$ is a $\mathbf{S}$-rooted c-forest in $\mathcal{G}'$, Lemma 1 implies that $Q[\mathbf{S}]$ is not identifiable from $\mathcal{G}'[\mathbf{A}_i']$.

Next result establishes the connection between non g-identifiability of $Q[\mathbf{S}]$ from $(\mathbb{A}, \mathcal{G})$ and non g-identifiability of $Q[\mathbf{S}]$ from $(\mathbb{A}', \mathcal{G}')$.

**Lemma 2.** *If $Q[\mathbf{S}]$ is not g-identifiable from $(\mathbb{A}', \mathcal{G}')$, then $Q[\mathbf{S}]$ is not g-identifiable from $(\mathbb{A}, \mathcal{G})$.*

To complete the proof using Lemma 2, it is enough to show that $Q[\mathbf{S}]$ is not g-identifiable from $(\mathbb{A}', \mathcal{G}')$.

**From g-identifiability to a system of linear equations:**
To show that $Q[\mathbf{S}]$ is not g-identifiable from $(\mathbb{A}', \mathcal{G}')$, we introduce two models in $\mathbb{M}^+(\mathcal{G}')$ such that equations (4) and (5) are satisfied. That is, $Q[\mathbf{S}]$ cannot be uniquely computed from $\{Q[\mathbf{A}'_i]\}_{i=0}^m$.

Note that to define a SEM $\mathcal{M}$ over a causal graph $\mathcal{G}'$, it suffices to define the domains $\mathfrak{X}_X$ and either the conditional distributions $P^{\mathcal{M}}(X|Pa_{\mathcal{G}'}(X))$ or the corresponding equation in the SEM for all $X \in \mathbf{V}' \cup \mathbf{U}'$, where $\mathbf{V}'$ and $\mathbf{U}'$ denote the observed and unobserved variables in $\mathcal{G}'$. We define the domains of all variables to be finite, i.e., $|\mathfrak{X}_X| < \infty$ for all $X \in \mathbf{V}' \cup \mathbf{U}'$. Let $U_0 \in \mathbf{U}'$ be a fixed unobserved variable (we will discuss later how to select $U_0$) with domain $\mathfrak{X}_{U_0} := \{\gamma_1, \cdots, \gamma_d\}$. We define both models $\mathcal{M}_1$ and $\mathcal{M}_2$ to have similar distributions over all variables except variable $U_0$ (We will specify these distributions in Section 5.) More specifically, for all $V \in \mathbf{V}'$,

$$P^{\mathcal{M}_1}(V \mid Pa_{\mathcal{G}'}(V)) = P^{\mathcal{M}_2}(V \mid Pa_{\mathcal{G}'}(V)), \quad (6)$$

and for all $U \in \mathbf{U}' \setminus \{U_0\}$,

$$P^{\mathcal{M}_1}(U) = P^{\mathcal{M}_2}(U) = \frac{1}{|\mathfrak{X}_U|}. \quad (7)$$

As the distributions in Equations (6) and (7) are the same for both models, for the sake of brevity, we drop the superscripts $\mathcal{M}_1$ and $\mathcal{M}_2$ from here on. For $j \in [1:d]$, We define $P^{\mathcal{M}_1}(U_0 = \gamma_j) = 1/d$ and $P^{\mathcal{M}_2}(U_0 = \gamma_j) = p_j$, where we will specify $\{p_j\}_{j=1}^d$ later such that $\mathcal{M}_2 \in \mathbb{M}^+(\mathcal{G}')$ and both Equations (4) and (5) hold.

For $\mathbf{v} \in \mathfrak{X}_{\mathbf{V}'}$, $i \in [0:m]$, and $j \in [1:d]$, we define

$$\theta_{i,j}(\mathbf{v}) := \sum_{\mathbf{U}' \setminus \{U_0\}} \prod_{X \in \mathbf{A}'_i} P(x \mid Pa_{\mathcal{G}'}(X)) \prod_{U \in \mathbf{U}' \setminus \{U_0\}} P(u),$$

$$\eta_j(\mathbf{v}) := \sum_{\mathbf{U}' \setminus \{U_0\}} \prod_{X \in \mathbf{S}} P(x \mid Pa_{\mathcal{G}'}(X)) \prod_{U \in \mathbf{U}' \setminus \{U_0\}} P(u),$$

where the index $j$ indicates that $U_0 = \gamma_j$ in the factorizations. Using these definitions, we can write $\{Q[\mathbf{A}'_i]\}_{i=0}^m$ and $Q[\mathbf{S}]$ for both models $\mathcal{M}_1$ and $\mathcal{M}_2$ as follows:

$$Q^{\mathcal{M}_1}[\mathbf{A}'_i](\mathbf{v}) = \sum_{j=1}^d \frac{1}{d}\theta_{i,j}(\mathbf{v}),$$

$$Q^{\mathcal{M}_2}[\mathbf{A}'_i](\mathbf{v}) = \sum_{j=1}^d p_j\theta_{i,j}(\mathbf{v}), \quad (8)$$

and

$$Q^{\mathcal{M}_1}[\mathbf{S}](\mathbf{v}) = \sum_{j=1}^d \frac{1}{d}\eta_j(\mathbf{v}),$$

$$Q^{\mathcal{M}_2}[\mathbf{S}](\mathbf{v}) = \sum_{j=1}^d p_j\eta_j(\mathbf{v}). \quad (9)$$

As we mentioned, we need to define $\{p_j\}_{j=1}^d$ such that $\mathcal{M}_2 \in \mathbb{M}^+(\mathcal{G}')$ and both Equations (4) and (5) hold. Substituting Equations (8) and (9) into (4) and (5) yield the following set of equations.

$$\sum_{j=1}^d (p_j - \frac{1}{d})\theta_{i,j}(\mathbf{v}) = 0, \quad \forall \mathbf{v} \in \mathfrak{X}_{\mathbf{V}'}, i \in [0, m],$$

$$\sum_{j=1}^d (p_j - \frac{1}{d})\eta_j(\mathbf{v}_0) \neq 0, \quad \exists \mathbf{v}_0 \in \mathfrak{X}_{\mathbf{V}'},$$

$$\sum_{j=1}^d p_j = 1, \quad (10)$$

$$0 < p_j < 1, \quad \forall j \in [1:d].$$

Note that the last inequalities ensure that $\mathcal{M}_2 \in \mathbb{M}^+(\mathcal{G}')$. The system of linear equations in (10) is solvable with respect to $\{p_j\}_{j=1}^d$ if and only if the following system of linear equations is solvable with respect to $\{\beta_j\}_{j=1}^d$.

$$\sum_{j=1}^d \beta_j\theta_{i,j}(\mathbf{v}) = 0, \quad \forall \mathbf{v} \in \mathfrak{X}_{\mathbf{V}'}, i \in [0:m]$$

$$\sum_{j=1}^d \beta_j\eta_j(\mathbf{v}_0) \neq 0, \quad \exists \mathbf{v}_0 \in \mathfrak{X}_{\mathbf{V}'} \quad (11)$$

$$\sum_{j=1}^d \beta_j = 0.$$

**Remark 3.** *If $\{\beta_j^*\}_{j=1}^d$ is a solution for (11), then*

$$p_j^* := \frac{1}{d} + \frac{\beta_j^*}{2hd}, \quad (12)$$

*is a solution for (10), where $h = \max_{j \in [1:d]} |\beta_j^*|$. Note that the division by $2h$ in Equation (12) ensures that $0 < p_j^* < 1$ for each $j \in [1:d]$.*

A solution to the system of linear equations in (11) will specify the distribution of $U_0$ in model $\mathcal{M}_2$. Clearly, existence of a solution to (11) depends on the choices of $\{\theta_{i,j}(\mathbf{v})\}$ and $\{\eta_j(\mathbf{v})\}$. The following result presents a sufficient condition under which (11) admits a solution.

For $\mathbf{v} \in \mathfrak{X}_{\mathbf{V}'}$ and $i \in [0:m]$, let $\theta_i(\mathbf{v})$ and $\eta(\mathbf{v})$ denote the vectors $(\theta_{i,1}(\mathbf{v}), ..., \theta_{i,d}(\mathbf{v}))$ and $(\eta_1(\mathbf{v}), ..., \eta_d(\mathbf{v}))$ in $\mathbb{R}^d$, respectively.

**Lemma 3.** *Consider the following set of vectors in $\mathbb{R}^d$*

$$\Omega := \{\theta_i(\mathbf{v}) : \ i \in [0:m], \mathbf{v} \in \mathfrak{X}_{\mathbf{V}'}\} \cup \{\mathbb{1}_d\}, \quad (13)$$

*where $\mathbb{1}_d$ denotes the all-ones vector in $\mathbb{R}^d$. If there exists $\mathbf{v}_0 \in \mathfrak{X}_{\mathbf{V}'}$ such that $\eta(\mathbf{v}_0)$ is linearly independent from all the vectors in $\Omega$, then the system of linear equations in (11) admits a solution.*

To summarize, so far, we have introduced two models for proving the necessary part of Theorem 1. In order to complete the proof, it remains to specify the conditional distributions in (6) for all observed variables which consequently specify the vectors in $\Omega$ in Equation (13) and to find a realization $\mathbf{v}_0 \in \mathfrak{X}_{\mathbf{V}'}$ such that $\eta(\mathbf{v}_0)$ is linearly independent from the set of the vectors in $\Omega$.

**Constructing the conditional distributions:** In order to specify the conditional distributions in (6), we first introduce the following definitions and notations.

Since $\mathbf{B}_0$ is a single c-component, the bidirected edges in $\mathcal{G}'_{\mathbf{B}_0}$ form a connected graph. Hence, there exists a bidirected edge between $\mathbf{S}$ and $\mathbf{B}_0 \setminus \mathbf{S}$. Accordingly, let $U_0$ be an unobserved variable in subgraph $\mathcal{F}_0$ that has one child in $\mathbf{S}$ and one child in $\mathbf{T} := \mathbf{V}' \setminus \mathbf{S}$. We denote the set of unobserved variables in $\mathcal{G}[\mathbf{S}]$ by $\mathbf{U}^{\mathbf{S}}$ and define $\mathbf{U}^{\mathbf{T}} := \mathbf{U}' \setminus (\mathbf{U}^{\mathbf{S}} \cup \{U_0\})$. For $X \in \mathbf{V}' \cup \mathbf{U}'$, we define $\alpha(X)$ to denote the number of graphs in $\{\mathcal{F}_j\}_{j=0}^k$ that contains $X$.

For each $i \in [0:k]$, let $T_i$ denotes a node in $\mathbf{B}_i \setminus \mathbf{S}$ such that $Ch_{\mathcal{F}_i}(T_i) \cap \mathbf{S} \neq \varnothing$. Note that such variables exist because $\mathcal{F}_i$s are $\mathbf{S}$-rooted c-forest.

Now, we are ready to introduce the domains of all variables in $\mathbf{V}' \cup \mathbf{U}'$. Recall that $\mathbf{V}' = \mathbf{S} \cup \mathbf{T}$ and $\mathbf{U}' = \mathbf{U}^{\mathbf{S}} \cup \mathbf{U}^{\mathbf{T}} \cup \{U_0\}$.

$$\begin{aligned}
\mathfrak{X}_X &:= [0:\kappa], \quad \forall X \in \mathbf{S}, \\
\mathfrak{X}_X &:= \{0,1\}^{\alpha(T)}, \quad \forall X \in \mathbf{T}, \\
\mathfrak{X}_X &:= [0:\kappa], \quad \forall X \in \mathbf{U}^{\mathbf{S}}, \\
\mathfrak{X}_X &:= \{0,1\}^{\alpha(U)}, \quad \forall X \in \mathbf{U}^{\mathbf{T}}, \\
\mathfrak{X}_{U_0} &:= [0:\kappa] \times \{0,1\}^{\alpha(U_0)-1}.
\end{aligned}$$

In the above definition, $\kappa$ is an arbitrary odd integer greater than 4. Note that the number of elements in $\mathfrak{X}_{U_0}$ is $d = (\kappa+1)2^{\alpha(U_0)-1}$.

According to the above definitions, for each $X \in \mathbf{T} \cup \mathbf{U}^{\mathbf{T}} \cup \{U_0\}$, its domain $\mathfrak{X}_X$ is a subset of $\mathbb{R}^{\alpha(X)}$ and it belongs to exactly $\alpha(X)$ number of subgraphs in $\{\mathcal{F}_i\}_{i=0}^k$. Suppose $X$ belongs to $\mathcal{F}_{i_1}, \cdots, \mathcal{F}_{i_{\alpha(X)}}$, where $i_1 < \cdots < i_{\alpha(X)}$. Thus, we denote $X$ by a vector $(X[i_1], \cdots, X[i_{\alpha(X)}])$ of length $\alpha(X)$. Next, we construct the conditional distributions of the observed variables by specifying their functional dependencies to their parents.

When $X \in \mathbf{T}$, we define the entries of its corresponding vector as

$$X[i_j] \equiv \left( \sum_{Y \in Pa_{\mathcal{F}_{i_j}}(X)} Y[i_j] \right) \pmod 2,$$

where $j \in [1 : \alpha(X)]$.

We now construct the variables in $\mathbf{S}$. Recall that $U_0$ has one child in $\mathbf{S}$ which we denote it by $S_0$. For each $S \in \mathbf{S} \setminus \{S_0\}$ and any realization of $Pa_{\mathcal{G}'}(S)$, we define $\mathbb{I}(S)$ to be one if there exists $i \in [0:k]$ such that

1. $T_i \in Pa_{\mathcal{G}'}(S)$ and $T_i[i] = 0$, or
2. there exists $X \in Pa_{\mathcal{G}'}(S) \setminus (\mathbf{U}^{\mathbf{S}} \cup \{T_i\})$ such that $\mathcal{F}_i$ contains $X$ and $X[i] = 1$,

and zero, otherwise. Note that according to the definition of $T_i$, it belongs to $\mathcal{F}_i$ and therefore, $T_i[i]$ exists. Analogously, we define $\mathbb{I}(S_0)$ to be one if there exists $i \in [0:k]$ such that

1. $T_i \in Pa_{\mathcal{G}'}(S)$ and $T_i[i] = 0$, or
2. $i \neq 0$, $\mathcal{F}_i$ contains $U_0$, and $U_0[i] = 1$, or
3. there exists $X \in Pa_{\mathcal{G}'}(S) \setminus (\mathbf{U}^{\mathbf{S}} \cup \{T_i, U_0\})$ such that $\mathcal{F}_i$ contains $X$ and $X[i] = 1$.

Now, for each $S \in \mathbf{S}$ and $s \in [0:\kappa]$, we define $P(S = s \mid Pa_{\mathcal{G}'}(S))$ as

$$\begin{cases}
\frac{1}{\kappa+1} & \text{if } \mathbb{I}(S) = 1 \\
1 - \kappa\epsilon & \text{if } \mathbb{I}(S) = 0 \text{ and } s \equiv M(S) \pmod{\kappa+1}, \\
\epsilon & \text{if } \mathbb{I}(S) = 0 \text{ and } s \not\equiv M(S) \pmod{\kappa+1},
\end{cases}$$

where $0 < \epsilon < \frac{1}{\kappa}$ and

$$M(S) := \begin{cases}
\sum_{x \in Pa_{\mathcal{G}'[\mathbf{S}]}(S)} x & \text{, if } S \in \mathbf{S} \setminus \{S_0\}, \\
u_0[0] + \sum_{x \in Pa_{\mathcal{G}'[\mathbf{S}]}(S)} x & \text{, if } S = S_0.
\end{cases}$$

Note that $M(S)$ is an integer number. This is because $Pa_{\mathcal{G}'[\mathbf{S}]}(S) \subseteq \mathbf{U}^{\mathbf{S}}$ and thus all terms in the above definition belong to $[0:\kappa]$.

**Lemma 4.** *The SEM constructed above belongs to $\mathbb{M}^+(\mathcal{G}')$.*

**Existence of realization $\mathbf{v}_0$:** Herein, we show that for the aforementioned conditional distributions, there exists a realization $\mathbf{v}_0$ such that $\eta(\mathbf{v}_0)$ is linearly independent from the set of the vectors in $\Omega$ (in Equation (13)). Consider the following subset of $\mathfrak{X}_{U_0} = \{\gamma_1, ..., \gamma_d\}$ with $\frac{\kappa+1}{2}$ elements:

$$\Gamma := \left\{ (2x, 0, \cdots, 0) : x \in [0 : \frac{\kappa-1}{2}] \right\}.$$

Recall that for $\mathbf{v} \in \mathfrak{X}_{\mathbf{V}'}$ and $i \in [0:m]$, $\theta_i(\mathbf{v})$ and $\eta(\mathbf{v})$ are two vectors in $\mathbb{R}^d$ with $j$-th entry corresponds to $U_0 = \gamma_j$. Suppose that $\Gamma = \{\gamma_{j_1}, ..., \gamma_{j_{\frac{\kappa+1}{2}}}\}$. Next result shows that in our constructed models, all entries of $\theta_i(\mathbf{v})$ with indices in $\{j_1, ..., j_{\frac{\kappa+1}{2}}\}$ are equal.

**Lemma 5.** *For any* $\mathbf{v} \in \mathfrak{X}_{\mathbf{V}'}$ *and* $i \in [0 : m]$,

$$\theta_{i,j_1}(\mathbf{v}) = \theta_{i,j_2}(\mathbf{v}) = \cdots = \theta_{i,j_{\frac{\kappa+1}{2}}}(\mathbf{v}).$$

An immediate consequence of this result is that any linear combination of the vectors in $\mathbf{\Omega}$ will have equal entries at the indices in $\{j_1, ..., j_{\frac{\kappa+1}{2}}\}$. Next, we show there exists a realization $\mathbf{v}_0$ for which $\eta(\mathbf{v}_0)$ does not follow this pattern and thus it is linearly independent of all vectors in $\mathbf{\Omega}$.

**Lemma 6.** *There exists* $0 < \epsilon < \frac{1}{\kappa}$ *for which there exists* $\mathbf{v}_0 \in \mathfrak{X}_{\mathbf{V}'}$ *and* $1 \le r < t \le \frac{\kappa+1}{2}$ *such that*

$$\eta_{j_r}(\mathbf{v}_0) \neq \eta_{j_t}(\mathbf{v}_0).$$

Lemma 6 implies that there exist $\mathcal{M}_1$ and $\epsilon$ for which there exists $\mathbf{v}_0 \in \mathfrak{X}_{\mathbf{V}'}$ such that $\eta(\mathbf{v}_0)$ is linearly independent from the set of vectors in $\mathbf{\Omega}$. As we discussed before, this completes our proof for Theorem 1.

# 6 CONCLUSION

We revisited the problem of general identifiability and showed that the positivity assumption of observational distributions is crucial for the soundness of do-calculus rules. This assumption was ignored in previous work. We presented a novel algorithm for g-identifiability, which is provably sound and complete considering the positivity assumption.

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
