# OpenReview forum: "Revisiting the General Identifiability Problem"
_auai.org/UAI/2022/Conference — UAI 2022 Oral_

### Official Review · Reviewer_xZJp · 2022-03-28

**Q2(1) Originality/Novelty:** 3
**Q2(2) Significance/Impact:** 3
**Q2(3) Correctness/Technical Quality:** 4
**Q2(6) Clarity Of Writing:** 4
**Q6 Overall Score:** 7
**Q8 Confidence In Your Score:** 5

**Q1 Summary And Contributions:**

This paper revisits the general identifiability problem of Lee et al. 2019. They note that this paper did not consider the positivity assumption carefully, which is a common condition in causal identification. Through a counterexample they demonstrate that it is possible for a causal effect in a non-positive distributions to be not identified. The original gID algorithm does not account for positivity, and uses a non-positive distribution in the proof of non-ID. The authors address these issues.

**Q10 Ethical Concerns (Optional):**

No.

**Q2 Assessment Of The Paper:**

More detailed information regarding each of these aspects is given below:

**Q2(5) Reproducibility:**

4: Excellent: Key resources (e.g., proofs, code, data) are available and key details (e.g., proof sketches, experimental setup) are comprehensively described for competent researchers to confidently and easily reproduce the main results.

**Q3 Main Strengths:**

The paper addresses the issue of positivity which has been less carefully considered in the graphical identification literature (compared with e.g. biostatistics and epidemiology causal literature, in which positivity assumptions are routinely stated). This paper highlights an issue with a high-impact paper in the literature and would be interesting to researchers in this sub-area.

Papers in identification theory can be verbose and full of jargon, but this paper is well-written. The authors are careful to serve the main ideas upfront with the minimum of equations, and they also provide a helpful sketch of the proof which is easy to follow (without having to read the theorems carefully).

The authors use proof techniques drawing on linear algebra ideas to provide the non-ID construction, which are novel and I believe of interest to the community. The proofs themselves are carefully done and appear to be correct.

**Q4 Main Weakness:**

One issue is that the new approach to the proof of completeness is not quite as general as the one it replaced. For example, the old completeness proof only required binary variables throughout (and so would be a valid construction for any discrete variable model since we can simply coarsen down to a binary variable). However, this new proof requires certain variables to have cardinality at least 5. It would appear that if we started with a binary variable model, this proof would not work and we would not know anything about completeness of gID queries on such models.

This first issue leads into my second issue, which is that I seek further clarity on why exactly ignoring positivity invalidates the gID proof. These comments are detailed further below.

The work has some small typos throughout, which will probably be spotted upon rereading the paper with fresh eyes.

**Q5 Detailed Comments To The Authors:**

As I understand the logical flow of the argument is as follows:

- gID admits models in M(G)
- the authors present an example which is a non-positive distribution in M(G), for which the causal effect is provably not identified.
- the authors point out that gID is not sound, and that it also uses non-positive distributions in its construction of the non-ID model
- the authors conclude that gID is flawed and the results are no longer valid
- furthermore the authors conclude it is better to implement gID admitting only models in M+(G)
- the proof of completeness must be redone, since the old proof relied on models in M(G) \ M+(G)

Now while this is perfectly valid I wonder if the authors considered alternative approaches to the argument. For one, it is quite possible to encounter non-positive distributions in practice. Rather than excluding these from the set of models, would it not make sense to include them and then modify when gID returns a result? I view this issue much in the same way as how M(G) is not defined excluding distributions that contain thickets obscuring identification - why should we treat these impediments to identification in different ways? Then the paper would look at when positivity impedes identification much in the same way that thickets do not always impede identification. The algorithm would need to be modified to be sound, and the completeness construction would now require a part for thickets and a part for positivity (so rather than a specific example of ID failure, the authors would then construct a general version of their example 2)

Another issue I see is that the logical jump between the third and fourth points in the above flow is not entirely clear, although the authors assert that it is. When identification fails in gID two models are constructed that have the same observed data distribution but correspond to different causal effects, and in these models the observed data distribution is not positive. Even pointing out the positivity issue, it is still the case that a construction has been presented that shows that no function exists to identify the causal effect, and so the proof still holds (note that at no point do we actually need to call the gID algorithm in the identification failure case). So what exactly is the issue with non-positive distributions in the completeness proof?

I can see that if we are to move to M+(G), then we ought to only consider models in that set throughout so that would be a fine motivation to move the completeness construction to M+(G). But this is somewhat subtle and I would encourage the authors to clearly state what exactly goes wrong, rather than asserting that moving to M+(G) is "crucial" since alternative paths exist. I think it's fine to move to M+(G) but readers should be aware that there are other ways to fix the positivity issue.





**Q7 Justification For Your Score:**

My score was based on the technical quality of the work. It is a clearly written paper, it solves an interesting problem (at least in its sub-area), and it introduces some new proof techniques that might be useful later on.

At the same time I do have some questions about the approach and some aspects of the proofs. I am more than willing to revise my overall score upon the author's thoughts on some of these issues.

**Q9 Complying With Reviewing Instructions:**

1: Yes.

---

### Official Review · Reviewer_J61W · 2022-03-30

**Q2(1) Originality/Novelty:** 3
**Q2(2) Significance/Impact:** 3
**Q2(3) Correctness/Technical Quality:** 3
**Q2(6) Clarity Of Writing:** 4
**Q6 Overall Score:** 7
**Q8 Confidence In Your Score:** 4

**Q1 Summary And Contributions:**

The problem of general causal effect identifiability (g-identifiability) is considered with the addition of the positivity assumption, and the authors show that previously proposed g-identifiability algorithm is not sound and complete in this context. The authors propose a new sound and complete algorithm under the positivity assumption.

**Q2 Assessment Of The Paper:**

More detailed information regarding each of these aspects is given below:

**Q2(5) Reproducibility:**

3: Good: Key resources (e.g., proofs, code, data) are available and key details (e.g., proofs, experimental setup) are sufficiently well-described for competent researchers to confidently reproduce the main results.

**Q3 Main Strengths:**

The paper is well motivated, and the fundamentally important assumption of positivity is heavily emphasized, and the authors clearly illustrate how the previously proposed g-identifiability algorithm fails in a simple scenario where the positivity assumption is not satisfied.

Fundamentals of causal models and identifiability theory are explained concisely.

A new sound and complete algorithm is provided under the positivity assumption, which has a surprisingly simple form, at least for those familiar with c-components and c-factors and the ID-algorithm.

**Q4 Main Weakness:**

The latter part of the paper starting from Section 5 provides the construction used to show non-identifiability is important, but very technical, and perhaps not that interesting on its own.

The positivity assumption could be explained in more practical terms, i.e., what does it mean when one has to actually estimate the causal quantity of interest.

The authors only consider the "full" positivity assumption (the joint distribution is positive for any value assignment of the observed variables), but the potential for relaxing this assumption for their algorithm is not discussed or considered.

**Q5 Detailed Comments To The Authors:**

In addition to Example 2 that shows how the previous g-identifiability algorithm erroneously identifies the causal effect, I think it would benefit the reader to also provide a running example showing how the new algorithm would proceed in the same scenario.

The practical implications of the positivity assumptions could be discussed. For example, if we consider a real data set with discrete variables, it may be the case that some combination of values never appears in the data, thus violating the positivity assumption if empirical probabilities are used in the estimation.

I also wonder if the authors considered whether it is possible to relax their assumption of positivity or does any relaxation necessarily result in an algorithm that is no longer sound and complete (perhaps a different form of relaxation than the one used by Shpiter and Pearl with the ID algorithm).

While this is not in the scope of the paper, I wonder whether the authors have considered possible implications of the positivity assumption on related work on g-transportability.

**Q7 Justification For Your Score:**

The authors' work provides in essence, a correction to a highly general causal effect identifiability algorithm. I see this as an important contribution.

The paper is very technical at some points, and I believe some additional details (such as the construction for the non-id case of the proof) could be moved to the appendix in favor of more examples on their algorithm and discussion on positivity.

**Q9 Complying With Reviewing Instructions:**

1: Yes.

---

### Official Review · Reviewer_nnFU · 2022-04-08

**Q2(1) Originality/Novelty:** 3
**Q2(2) Significance/Impact:** 3
**Q2(3) Correctness/Technical Quality:** 3
**Q2(6) Clarity Of Writing:** 3
**Q6 Overall Score:** 7
**Q8 Confidence In Your Score:** 2

**Q1 Summary And Contributions:**

The paper provides a (in my view) very important correction to the g-identifiability which has previously been proposed.
The correction is based on the overlooked issue of the need to assume positivity of the observational distribution, i.e. that all combinations that can be thought of under interventions, can possibly be observed in the data.
With this correction, a new algorithm (#2) is provided here, and its soundness & completeness is shown.

**Q2 Assessment Of The Paper:**

More detailed information regarding each of these aspects is given below:

**Q2(4) Quality Of Experiments (Optional):**

3: Good: The experimental evaluation is adequate, and the results convincingly support the main claims.

**Q2(5) Reproducibility:**

4: Excellent: Key resources (e.g., proofs, code, data) are available and key details (e.g., proof sketches, experimental setup) are comprehensively described for competent researchers to confidently and easily reproduce the main results.

**Q3 Main Strengths:**

g-identifiability is an important concept for the field of causality and it is therefore important to get it right; certainly in practice the issue of positive often gets overlooked; the correction here seems very pertinent and a new identification algorithm is provided.
It seems to me that such an assumption of positivity is quite fundamental to many aspects of causal inference, and it would indeed be suspicious and need justification if a methods claims that it can do without this assumption.


**Q4 Main Weakness:**

the paper is hard to read, after a nice introduction it mainly consists of the proof
the topic is, however, very technical by nature, so it is admittedly hard to write a nice readable paper about it.
I suggest below some points where maybe the authors could add a bit more on the intuition (but I am aware that not all may apply or be realistic)



**Q5 Detailed Comments To The Authors:**

it would be nice to get an idea / more intuition of the following points
- I think more could be done to illustrate and give intuition behind the role and importance of the positivity assumption,
- is the role of positivity different for g-identifiability than for ID itself?
- in what key aspects of the inner working is the new algorithm different from the Lee_etal-algorithm?
- how does this positivity assumption relate to the "usual" positivitiy assumption underlying the estimation of causal effects?
- also some readers might need more convinving that this is an important correction and does not just concern degenerate / unrealistic cases
- it is also unclear whether the cases where Lee et al's algorithm fail have "positive probability" or will essentially not occur.
- what is the relation to the identification of cross-world quantities (e.g. certain path-sepcific effects) where positivity will never hold (and experimentation is not possible)?
- under Q2(4) we were asked about the experimental evaluation which does not apply here; instead one could think about the quality of the examples: is there a way to see that counterexamples to Lee_etal are easy to construct? (this links with my above comments)


minor:
there are many language problems with missing "the" and plural-"s" and strange grammatical constructs etc. I cannot possibly list all of them here, but the authors should go through the text carefully

**Q7 Justification For Your Score:**

I would like to see the result published for the above reasons; and I think the positivity assumption is not given enough attention; in an ideal world, the presentation would be a bit more readable.

**Q9 Complying With Reviewing Instructions:**

1: Yes.

---

### Official Review · Reviewer_GjSV · 2022-04-12

**Q2(1) Originality/Novelty:** 3
**Q2(2) Significance/Impact:** 2
**Q2(3) Correctness/Technical Quality:** 3
**Q2(6) Clarity Of Writing:** 4
**Q6 Overall Score:** 7
**Q8 Confidence In Your Score:** 3

**Q1 Summary And Contributions:**

The authors show that the algorithm for general identifiability of Lee et al. 2019 is not sound. The authors provide a counterexample and show that the algorithm is not sound. Moreover, they show that the completeness proof of the algorithm of Lee et al. 2019 only is fulfilled if you ignore the positivity assumption. The main contribution of this paper is that they provide a new algorithm for general identifiability that is both sound and complete under the positivity assumption.


**Q2 Assessment Of The Paper:**

More detailed information regarding each of these aspects is given below:

**Q2(5) Reproducibility:**

4: Excellent: Key resources (e.g., proofs, code, data) are available and key details (e.g., proof sketches, experimental setup) are comprehensively described for competent researchers to confidently and easily reproduce the main results.

**Q3 Main Strengths:**

The paper addresses a crucial flaw in the proof of the soundness of the algorithm of the general identifiability problem of Lee et al. 2019. They provide an insightful counterexample and offer a new algorithm with the interesting property that it decomposes the general identifiability problem into a series of classical identifiability sub-problems. In my opinion, the paper provides a significant result that solves an issue of which the community may not be aware.

Overall, the paper is very well written and organized. It gives an excellent exposition of the general idea, making it a pleasure to read. The supplementary material supports all theoretical claims well; however, I did not check all the proofs there, to be honest.


**Q4 Main Weakness:**

The fact that the proofs of the main theoretical results are contained in the supplementary material, makes this paper less self-contained. Although this is the only slight weakness of the paper since the overall readability is quite reasonable given the page limit. There were some small parts that were difficult to follow or where the readability could be improved (see detailed comments below).



**Q5 Detailed Comments To The Authors:**

Some parts that were difficult to follow:
- p.7: "Note that such variable exists ..." <-- Not clear why?
- p.8: "\kappa is an arbitrary odd integer greater than 4" <-- Why is that?

Readability could be improved in section 5:
- For example, before the paragraph "For sake of space, ..." on page 6, it would help if the authors explained that it suffices here to construct M_1 and M_2 (which is what the rest of section 5 is all about),
- Or right before the conclusion in section 6 it would help to summarize that the last lemma is the final piece of the puzzle to construct M_1 and M_2 and that they have the right properties that are needed to prove the Theorem.

Minor comments:
- p.1: "There are two main assumptions ... " <-- Are these assumptions or are these inputs to the problem?
- p.7: In "Next result presents a necessary condition ..." <-- Shouldn't this be replaced by "a sufficient condition"?


**Q7 Justification For Your Score:**

This is a good paper that is very well written. Under the condition that all the proofs are correct (which I didn't all check), I would give this paper an accept.


**Q9 Complying With Reviewing Instructions:**

1: Yes.

---

### Decision · Program_Chairs · 2022-05-15

**Decision:**

Accept (Oral)

**Comment:**

Meta Review: In this paper, the authors revisit recent work (Lee et al., 2019) on identification of interventional distributions from a set that may include the observed data distribution and/or a set of interventional distributions.  (Lee et al., 2019) proposed a sound and complete identification algorithm for that problem.  The authors point out a critical flaw in that work: the completeness proof relies in a crucial way on absence of positivity.  However, absence of positivity (even including relaxations of positivity proposed by Shpitser and Pearl) results in the proposed algorithm not being sound, as the authors convincingly illustrate with examples.  The authors provide a repair of the algorithm proposed by (Lee et al., 2019), with new soundness and completeness proofs.

The reviewers were unanimously positive about this paper, and their questions were answered by the authors to their satisfaction.  Some reviewers pointed out that in addition to repairing a critical flaw in a recent paper, this work will also encourage authors in the causal graphical modeling community to pay more careful attention to important issues related to positivity or "overlap" assumptions.